# Theoretical Investigation of Near-Infrared Fabry–Pérot Microcavity Graphene/Silicon Schottky Photodetectors Based on Double Silicon on Insulator Substrates

**DOI:** 10.3390/mi11080708

**Published:** 2020-07-22

**Authors:** Maurizio Casalino

**Affiliations:** Institute of Applied Science and Intelligent Systems “Eduardo Caianiello” (CNR), Via P. Castellino n. 141, 80131 Naples, Italy; maurizio.casalino@na.isasi.cnr.it

**Keywords:** resonant cavity, photodetectors, near-infrared, silicon, graphene

## Abstract

In this work a new concept of silicon resonant cavity enhanced photodetector working at 1550 nm has been theoretically investigated. The absorption mechanism is based on the internal photoemission effect through a graphene/silicon Schottky junction incorporated into a silicon-based Fabry–Pérot optical microcavity whose input mirror is constituted by a double silicon-on-insulator substrate. As output mirror we have investigated two options: a distributed Bragg reflector constituted by some periods of silicon nitride/hydrogenated amorphous silicon and a metallic gold reflector. In addition, we have investigated and compared two configurations: one where the current is collected in the transverse direction with respect to the direction of the incident light, the other where it is collected in the longitudinal direction. We show that while the former configuration is characterized by a better responsivity, spectral selectivity and noise equivalent power, the latter configuration is superior in terms of bandwidth and responsivity × bandwidth product. Our results show responsivity of 0.24 A/W, bandwidth in GHz regime, noise equivalent power of 0.6 nW/cm√Hz and full with at half maximum of 8.5 nm. The whole structure has been designed to be compatible with silicon technology.

## 1. Introduction

Silicon (Si) photonics is nowadays an emerging market promising to reach a value of $560 M at chip level and $4 B at transceiver level in 2025 as shown in Figure 1. Indeed, both switching and interconnects of the existing data center risk becoming an early bottleneck for the huge increase in internet data traffic driven by social network and video contents. 

This is because in the future new technologies must necessarily be introduced for making Si fully compatible for sensors [2,3] and photonic devices in general. Since the 1980s the interest of both scientists and industry in Si photonics has grown exponentially. Nowadays Luxtera together with Intel share the leadership in Si photonics with commercial transceivers able to transmit data at a rate of 100 G. Si is a very mature technology and still plays a key role in the microelectronic industry, for this reason the realization of photonic devices in Si would be the best approach for matching the data center requirements in terms of reliability, low cost, power consumption and integration density. 

Photodetectors (PDs) are key devices in photonics making possible the transduction of light into current. Si PDs operating in the visible spectrum are still commercial components, on the other hand Si employment for near-infrared (NIR) detection is hindered by its optical transparence over 1.1 micron. 

Currently, Si-based NIR PDs take advantage of the integration of germanium (Ge) [4,5], however, these devices are characterized by high leakage current due to the lattice mismatch of 4.3%. In order to mitigate this drawback a buffer layer, gradually matching the Si to the Ge lattice, can be fabricated [5,6,7]; even if this approach reduces the leakage current, however it remains quite high. In addition, the fabrication of this buffer layer involves high thermal budget fabrication processes [8] which prevent Ge being monolithically integrated on Si. Finally, the low Ge absorption at 1550 nm (one order of magnitude lower than indium gallium arsenide, InGaAs) hinders the realization of high-speed PIN devices due to the high thickness of the intrinsic region.

Internal photoemission effect (IPE) offers one option to the all-Si approach in the field of the NIR detection. IPE concerns the optical absorption of a metal involved in a Schottky junction and then the emission of the photo-excited carriers into the semiconductor over the Schottky junction [9,10]. Both palladium silicide (Pd_2_Si) and platinum silicide (PtSi) have been widely employed in infrared charge-coupled device (CCD) image sensors but, unfortunately, they have to work at cryogenic temperature for increasing the signal-to-noise ratio at an acceptable level. Pd_2_Si/Si Schottky PDs can operate in the spectrum ranging from 1 to 2.4 μm requiring temperatures of 120 K [11,12] (e.g., satellite applications) while PtSi/Si Shottky PDs can work to an extended spectrum ranging from 3 to 5 μm [13,14] but they needs to operate at lower temperatures of only 80 K. Focal plane arrays (FPA) based on 512 × 512 PtSi/Si PDs have been demonstrated [15]. 

In 2006, for first time, it was theoretically proposed to use IPE for detecting NIR in Si at room temperature. The idea was to work with junctions characterized by higher Schottky barriers for reducing the dark current and, at the same time, to recovery the unavoidable reduced efficiency by incorporating the junction into a Fabry–Peròt optical microcavity [16]. After that, many innovative approaches have been investigated taking advantage of: Si nanoparticles (NPs) [17], surface plasmon polaritons (SPPs) [18,19], antennas [20] and gratings [21]. Despite this, to date responsivities of only 30 mA/W [17] and 5 mA/W [22] were reported for waveguide and free-space PDs, respectively. The low responsivity is mainly due to the low emission probability of the charge carriers excited from the metal to the semiconductor. IPE theory shows that this emission probability can be increased by thinning the metal [23,24]. This is the reason why the idea of replacing metal with graphene was born. Graphene/Si Schottky PDs have shown unexpected efficiency in both the visible [25,26] and NIR [27] spectrum: while in the visible range this enhancement has been ascribed to the gating effect of the graphene/SiO_2_/Si capacitor in parallel to the graphene/Si junction [25,28], in the NIR range increased IPE has been attributed to the increased charge emission probability due to the mediation of the interface defects [27]. However, in this last case the whole efficiency is hindered by the low graphene absorption (only 2.3%). In order to increase the graphene optical absorption, many strategies have been followed: by realizing plasmonic nanostructures [29], by reducing the graphene size down to nanodisks [30] or quantum dots [31]. On the other hand, PDs based on the increase of thin film optical absorption by the use of an optical microcavity have been already reported in literature with the name of resonant cavity enhanced (RCE) photodetectors [32]. RCE PDs are able to shrink the optical field inside the cavity within the active intrinsic layer of III-V PIN diodes [32] allowing reducing the size of the absorbing layer, and consequently the carrier transit time, without degradation of the device efficiency. 

Taking advantage of this idea, in this work we propose a new concept of Si-based RCE PDs operating at 1550 nm where graphene/Si Schottky junctions have been incorporated into a Fabry–Perot microcavity [33] that could be realized starting from a crystalline-Si (c-Si) based distributed Bragg reflector (DBR) substrate. Indeed, thanks to a double silicon on insulator (SOI) process, DBRs consisting of two periods of c-Si/SiO_2_ can be realized and optimized for high reflectivity around 1550 nm [34]; they are named double-SOI (DSOI). As second mirror two options have been considered: a distributed Bragg reflector constituted by some periods of silicon nitride (Si_3_N_4_)/hydrogenated amorphous silicon (a-Si:H) and a metallic reflector based on a thick layer of gold. In more detail, our proposal is to replace the III-V P-I-N diodes used in classical RCE configuration by a buffer layer added to a graphene/silicon Schottky junction. The buffer layer is useful to accommodate the localized optical field on the thin graphene layers where charge carriers are excited by photons and then emitted into c-Si through the Schottky junction making detection possible in the NIR spectrum. As buffer layer, we have chosen a-Si:H, a material which can be deposited at low temperature by a plasma-enhanced chemical vapor deposition (PECVD) system. Moreover, a-Si:H is characterized by a refractive index very close to that of c-Si at 1550 nm which is mandatory to reduce the Fresnel reflection at the interface and, consequently, to consider the a-Si:H/graphene/c-Si three-layer structure as one unique optical cavity. Finally, we have investigated the possibility to collect the current both longitudinally and transversally to the direction of the incoming light putting in evidence advantages and disadvantages of each configuration. 

## 2. Photodetector Performance and Theoretical Background

It is well-known that a very important figure of merit for a PD is the internal quantum efficiency (IQE) *η_int_*, defined as the number of charge carriers collected per absorbed photon. IQE is linked to the external quantum efficiency (EQE) *η_ext_* (number of charge carriers collected per incident photon) by the following formula: *η_ext_* = *A η_int_*, where *A* is the optical absorption of the active material. A macroscopic measurable magnitude is the responsivity *R*, i.e., the ratio of the photogenerated current (*I_ph_*) to the incident optical power (*P_inc_*). The responsivity *R* is linked to EQE *η_ext_* by the following formula:(1)$$R=\frac{I_{ph}}{P_{inc}}=\frac{\lambda (nm)}{1242}\eta_{ext}=\frac{\lambda (nm)}{1242}A\;\eta_{int}$$

As reported in literature, the internal quantum efficiency of an IPE-based graphene/silicon Schottky photodetectors is given by the following [35]:(2)$$\eta_{int}=\frac{1}{2}\frac{{\left(h\nu \right)}^{2}-{\left(q\phi_{B}\right)}^{2}}{{\left(h\nu \right)}^{2}}$$
where *hν* is the photon energy, *q* = 1.602 × 10^−19^ C is the charge electron and *qϕ_B_* is the Schottky barrier height of the graphene/silicon junction. It is well-known that due to the Fermi level shift in graphene, the Schottky barrier of a graphene/silicon junction lowers by increasing the reverse voltage applied. Of course, the decrease in *ϕ_B_* leads to increased responsivity, thus the effects of the reverse voltage *V_R_* on the responsivity can be investigated. In other words, the Schottky barrier *qϕ_B_* can be viewed as the sum of the Schottky barrier at zero-bias *qϕ_B_*_0_ and the Schottky barrier lowering *q*Δ*ϕ_B_(V_R_)* due to the increase in reverse voltage: *qϕ_B_ (V_R_)* = *qϕ_B_*_0_ + *q*Δ*ϕ_B_ (V_R_)*. To this aim we take advantage of the work of Tongay et al. [36] where, the Schottky barrier lowering Δ*ϕ_B_ (V_R_)* due to the Fermi level shift has been calculated by the following formula:(3)$$q\Delta \phi_{B}\left(V_{R}\right)=-\frac{1}{2}\hbar v_{F}\sqrt{\frac{\pi \varepsilon_{s}\varepsilon_{0}N\left(V_{bi}+V_{R}\right)}{2qn_{0}}}$$
where *n*_0_ is the graphene extrinsic doping (a typical value is 5 × 10^12^ cm^−2^ [36]), *N* is the semiconductor doping, *v_F_* = 1.1 × 10^8^ cm/s is the Fermi velocity, $\hbar =6.5\times 10^{-16}\;eVs$ is the Plank constant, *ε*_0_ = 8.859 × 10^−14^ C/cmV is the permittivity of vacuum, *ε_S_* = 11.4 is the relative permittivity of Si and *V_bi_* is the built-in potential of the junction.

Device bandwidth is another figure of merit very important for a PD, in particular in telecom and datacom applications. The main factors limiting the time response of a PD integrated into an optical microcavity are [37,38]: (1) the carriers transit time *τ_tr_* across the charge spatial region, (2) the charge/discharge time *τ_RC_* linked to both the junction capacitance *C_j_* and the load resistance *R_L_*; (3) the cavity photon lifetime *τ_ph_* [39]. Thus, the overall time constant of the PD is *τ* = *τ_tr_* + *τ_RC_* + *τ_ph_*. For the τ_RC_ = *R_L_C_j_* calculation, we can consider *R_L_* = 50 Ω (typical value for high-speed applications) and the junction capacitance given by the following formula: *C_j_* = *A_PD_*·*ε*_0_·*ε_s_*/W, where *A_PD_* = *πr*^2^ is the circular graphene area with radius r in contact with Si and W is the charge spatial region width On the other hand, the cavity photon lifetime can be calculated by the following formula *τ_ph_* = 1/2*πδν* [39], being *δν* the spectral width of the absorption peak at half maximum. The transit time *τ_tr_* can be written as *τ_tr_* = *t*/*v_sat_* where *t* is the maximum distance that the electron must travel before being collected (by considering this distance completely depleted) and *v_sat_* is the carrier saturation velocity in Si. As we will see, this distance t strongly affects the bandwidth of the device.

Finally, the cut-off frequency can be estimated as:(4)$$f_{3dB}=\frac{1}{2\pi \tau }=\frac{1}{\;2\pi \left(\frac{t}{v_{sat}}+\frac{\pi \varepsilon_{0}\varepsilon_{s}R_{L}}{W}r^{2}+\frac{1}{2\pi \delta \nu }\right)}$$

Another very important figure of merit is the noise equivalent power (NEP), i.e., the minimum optical power which can be detected by a PD, in an approximated form, which can be written as:(5)$$NEP=\frac{\sqrt{2qJ_{d}}}{R}$$

Being *q* the electron charge and *J_d_* the dark current density that, for Schottky PD, can be written as:(6)$$J_{d}=A^{*}T^{2}e^{-\frac{q\phi_{B0}}{kT}}$$
where *A** is the Richardson constant (32 A/cm^2^K^2^ for *p*-Si [37]), *T* the absolute temperature, *k* the Boltzmann constant and *qΦ_B_*_0_ is the Schottky barrier of the graphene/silicon Schottky junction at zero bias. The unity of measure of NEP is $W/cm\sqrt{Hz}$.

## 3. Photodetector Concept: From the Idea to the Device

In this section the basic idea of the device will be presented, some details on the numerical simulations will be provided, and the materials selected for a possible fabrication will be discussed. 

### 3.1. Si-Based Resonant Cavity Enhanced (RCE) Photodetectors

Classical RCE PDs are able to concentrate the enhanced optical field in the absorbing intrinsic region of a PIN diode realized by a III-V semiconductor [32], where the P, I and N are characterized by a slightly different stoichiometry in such a way as to neglect the reflections at the interface and to consider the whole PIN structure as an unique cavity. 

In our proposed device, the P-I-N structure has been replaced by a Schottky graphene/c-Si Schottky junction on which is added a buffer layer useful to accommodate the localized optical field on graphene as shown in Figure 2. As buffer layer we have chosen hydrogenated amorphous silicon (a-Si:H) because it can both be deposited at low temperature by a PECVD system and its refractive index at 1550 nm is 3.58 [40], very close to that one of c-Si (3.48) [41]. This latter property together with the high transparence of the graphene layer allow considering the whole a-Si:H/graphene/c-Si three-layer structure as one unique optical cavity with negligible reflections at the interfaces. 

It is widely reported in literature that the maximum absorption of RCE PDs occurs when the reflectivity of the output mirror is close to the unity. This is the reason why we consider illumination from the bottom as shown in Figure 2, indeed as will see in Section 3.3 the DSOI DBR is characterized by a limited reflectivity making it not suitable to work as output mirror. It is worth mentioning that an optimized RCE structure is characterized by an output mirror reflectivity *R*_2_ = 1 and an input mirror reflectivity $R_{1}=R_{2}\times e^{-\alpha_{g}d_{g}}$, being *α_g_* and *d_g_* the absorption coefficient and thickness of the graphene absorbing layer, respectively [35]. All numerical simulations have been carried out by the transfer matrix method (TMM) taking into account the dispersion curve shown in Figure 3a–d.

Figure 3b shows that only graphene and Au are provided of a non-negligible extinction coefficient (absorption coefficient) in the range of wavelength taken into account. Moreover, Figure 3d shows that non-negligible absorption appears when Si is considered heavily doped due to free carrier absorption. In the next, for heavily and lightly doped silicon semiconductor we’ll intend doping of 5 × 10^19^ cm^−3^ and 1 × 10^15^ cm^−3^, respectively.

All dispersion curves shown in Figure 3 have been taken by references [41,42,43], while the graphene complex refractive index *n_g_* can be obtained by [44]: (7)$$n_{g}=\sqrt{\varepsilon_{g}}=\sqrt{2.148+j\frac{G\lambda }{2d_{g}}}$$
where *ε_g_* is the relative permittivity of graphene, λ is the wavelength, *d_g_* = 0.335 nm is the graphene thickness and $G=\frac{q^{2}}{2\varepsilon_{0}hc}=0.0073$ is the fine structure constant [45] (being h = 6.626 × 10^−34^ Js the Planck constant, and c = 3 × 10^10^ cm/s the speed of light in vacuum).

Finally, the heavily-doped c-Si complex refractive index has been calculated by applying the theory of the free carrier absorption [46]. The calculation provides the variation of both the real part of the refractive index Δ*n_Si_* and the absorption coefficient Δα_Si_ of doped c-Si as a function of the donor and acceptor concentration atoms, *N_d_* and *N_a_*, respectively [46]:(8)$$\Delta n_{Si}=\frac{q^{2}\lambda^{2}}{8\pi^{2}c^{2}\varepsilon_{0}n_{Si}^{0}}\left(\frac{N_{d}}{m_{e}^{*}}+\frac{N_{a}}{m_{h}^{*}}\right)$$
(9)$$\Delta \alpha_{Si\;}=\frac{q^{3}\lambda^{2}}{4\pi^{2}c^{3}\varepsilon_{0}n_{Si}^{0}}\left(\frac{N_{d}}{\mu_{e}{\left(m_{e}^{*}\right)}^{2}}+\frac{N_{a}}{\mu_{h}{\left(m_{h}^{*}\right)}^{2}}\right)$$
where *n_Si_*^0^ is the refractive index of unperturbed crystalline Si, $m_{e}^{*}$ and $m_{h}^{*}$ are the conductivity effective mass of electrons and holes, respectively, while *μ_e_* and *μ_h_* are the electrons and holes mobility, respectively. It is worth remembering that the extinction coefficient *κ_Si_* can be derived by the absorption coefficient *α_Si_* by the following formula: *α_Si_* = *(4π/λ)*
*κ_Si_*. 

### 3.2. Buffer Layer

By moving our attention on the cavity, as already mentioned, we need to choose a material working as buffer layer which can deposited on graphene. The a-Si:H has been already successfully deposited on graphene without damaging it [47], in addition this material is characterized by a refractive index very close to that of c-Si at 1550 nm. This property combined with the high transparency of graphene lead to negligible Fresnel reflection at the a-Si:H/graphene and c-Si/graphene interfaces. Indeed, in literature has been already proved that the a-Si:H/graphene/c-Si three-layer cavity can be modelled by the classical theory of RCE PDs [40].

Figure 4 shows the reflections at the c-Si/graphene and a-Si:H/graphene interfaces. Low reflections in the order of 10^−4^ are reported at 1550 nm.

### 3.3. Output and Input Mirror of the Fabry–Pérot Microcavity

As output mirror of the Fabry-Pérot microcavity we take into account two options: a DBR constituted by alternating layer of a-Si:H/Si_3_N_4_ (that can be deposited at low temperature by a PECVD system) and a metal reflector (MR) constituted by a thick Au metallic layer. It is well-known that a DBR consists of a multilayer-stack of alternate high- and low-refractive index layers, all one quarter wavelength thick. In order to get high reflectivity at 1550 nm by a Si_3_N_4_/a-Si:H DBR the thicknesses are calculated as high as 213 and 108 nm, respectively. Figure 5a shows the DBR reflectivity for 3, 4 and 5 pairs of Si_3_N_4_/a-Si:H.

In addition, Figure 5a shows the reflectivity of a 200 nm-thick Au layer. It is well-known that metals are characterized by a plasma frequency higher than frequency in the optical spectrum, this inhibits the optical propagation in the metal leading to high reflectivity. 

Moving our attention to the input mirror, the possibility to fabricate a DBR by alternating layers of silicon dioxide (SiO_2_) and c-Si is reported in reference [34]. DBR constituted by alternating layers of c-Si/SiO_2_ are characterized by a large refractive index contrast (3.48/1.47 at 1550 nm, respectively) allowing the realization of high-reflectivity, wide spectral stop-band DBR made of few periods [48]. These DBRs are realized by a double silicon on insulator process, thus they are constituted by two c-Si/SiO_2_ pairs and named DSOI. Because the manufacturing process typically does not allow the manufacture of Si thickness as thin as λ/4n_Si_, a c-Si thickness of 3λ/4n is typically used [48]. A further advantage of this structure is that on top of the reflector there is a crystalline layer of Si which can be used for growing (by epitaxial processes) other crystalline Si layers with different doping, for instance for realizing heavily doped layers necessary for the fabrication of Ohmic contacts. We have investigated c-Si/SiO_2_ DSOI with thicknesses of 340 nm/270 nm in two configurations: one with the first c-Si layer heavily doped (HD) and the other with the first c-Si layer lightly doped (LD). We name them DSOI-HD and DSOI-LD, respectively. For the DSOI reflectivity calculation, c-Si has been considered as both input and output semi-infinite medium. In Table 1, reflectivity and thicknesses of all reflectors discussed in this section are reported. 

## 4. Results

This section will show the results of the numerical simulations carried out by TMM [49] implemented by custom codes written in Matlab. Devices can be realized in two configurations as shown in Figure 6a,b. 

In particular, Figure 6a shows a structure where the first layer of the DSOI is lightly doped while only a small region placed under the collecting metal is heavily doped for getting an Ohmic contact (DSOI-LD). In other words, in this configuration we can say that both Ohmic and Schottky contacts are realized on the same plane and photoexcited charge carriers emitted by graphene into c-Si are collected transversally to the direction of the incoming light. We name this configuration: the transverse collection device. As should be noted, in this configuration the maximum distance t that a charge carrier generated in the center of the graphene disk has to cover before being collected is further high because the radius of the graphene area is in the order of some tens of μm. As consequence, even if all this distance t is completely depleted the slow carrier transit time *τ_tr_* is expected to reduce the device bandwidth.

On the other hand, in Figure 6b is shown a structure where the first layer of the DSOI is entirely heavily doped (DSOI-HD). In other words, in this configuration the Ohmic contact is placed in front of the Schottky contact and the photoexcited charge carriers emitted by graphene into c-Si are collected in parallel (longitudinally) to the direction of the incoming light. We name this configuration: longitudinal collection device. It should be noted in this configuration that the distance t that any charge carrier emitted by graphene into c-Si has to cover before being collected is the thickness of the c-Si layer composing the cavity. This value is in the order of some hundreds of nm, as a consequence the carrier transit time is two orders of magnitude lower with respect to transverse collection devices shown in Figure 6a. However, the heavily doped layer in the DSOI reflector absorbs part of the light trapped in the cavity at any round-trip, thus in this configuration a reduced graphene absorption, and consequently responsivity, is expected. 

### 4.1. Transverse Collection Configuration (TCC)

In this section we investigate the transverse collection configuration (TCC) shown in Figure 6a. As output mirror, a DBR constituted of 3, 4 and 5 Si_3_N_4_/a-Si:H pairs is considered. On the other hand, the input mirror of the device is constituted by a DSOI-LD. Of course, as shown in Figure 6a, the optical microcavity is formed by a-Si:H/graphene/c-Si three-layer structure. 

We have performed numerical simulations in order to calculate the graphene absorption at 1550 nm by varying the thicknesses of both a-Si:H and c-Si layers comprising the cavity for a DBR output mirror composed of 3, 4 and 5 Si_3_N_4_/a-Si:H pairs. Results are shown in Figure 7a–c, respectively; because the position of the maximum of the standing wave inside the cavity does not depend on the reflectivity of two mirrors, in any case that the maximum graphene absorption can be obtained for 111 nm-thick and 214 nm-thick of c-Si and a-Si:H, respectively. 

The spectral graphene absorption around 1550 nm for the optimized thicknesses is shown in Figure 7d. Figure 7d shows that the maximum graphene absorption is 0.44, 0.54 and 0.58 while the full width at half maximum (FWHM) are 10.17 nm, 9 nm and 8.54 nm, for DBRs composed by 3, 4 and 5 Si_3_N_4_/a-Si:H pairs, respectively. Of course the maximum absorption is obtained for the cavity characterized by the highest finesse, i.e., that one provided of a DBR constituted by 5 pairs of Si_3_N_4_/a-Si:H. 

Taking advantage of the calculated graphene absorption, by applying Equations (1) and (2), we have calculated the spectral responsivity at zero bias.

As shown in Figure 8a the maximum responsivity at 1550 nm is 0.19 A/W, 0.23 A/W and 0.24 A/W for DBR composed of 3, 4 and 5 Si_3_N_4_/a-Si:H pairs, respectively.

In order to verify if a further increase in responsivity at 1550 nm can be obtained by increasing the reverse bias, we use Equations (1)–(3). Figure 8b shows a very limited increase in responsivity also at −10 V of reverse bias applied, leading to the idea that these devices could also work at low reverse voltage without degrading their efficiency. 

Moving our attention on the bandwidth of the device, Figure 9a–c show the time constants discussed in the Section 2 as function of the radius r of the graphene active area, for DBR composed by 3, 4 and 5 Si_3_N_4_/a-Si:H pairs, respectively. In addition, Figure 9a–c show the 3 dB roll-off frequency as function of the graphene disk radius r, too. 

Figure 9a–c have been calculated by considering: (i) for the *τ_tr_* calculation, a *v_sat_* = 10^7^ cm/s [37] and a drift length *t* = *r*; (ii) for the *τ_RC_* = *R_L_C_j_* calculation a load resistance *R_L_* = 50 Ω and a junction capacity *C_j_* = (*πr*^2^*ε*_0_*ε_s_*)/*W*, being *W* = √((2·*ε*_0_·*ε_s_*)/qN_a_)·*V_bi_* = 0.5 μm the length of the depletion layer that has been evaluated by considering a built-in potential *V_bi_* = *Φ*_B0_ − (*E_F_* − *E_V_*) = 0.196 V (with the Schottky barrier *Φ*_B0_ = 0.45 V [50] and the difference between the extrinsic Fermi level and the Si valence band *E_F_* − *E_V_* = 0.254 V calculated starting from a *p*-type doping *N*_a_ = 10^15^ cm^−3^; (iii) the cavity photon lifetime *τ_ph_* = 1/2π*δν*, being *δν* the spectral width of the absorption peak which can be obtained by the FWHM extracted by Figure 7d and converted into frequencies leading to: *δν* = 1270, 1124 and 1066 GHz for DBR composed by 3, 4 and 5 Si_3_N_4_/a-Si:H pairs, respectively. 

Figure 9a–c show that in this configuration the limiting factor is the transit time; of course, by increasing the radius r, the τ_RC_ constant time grows in a square way approximating the value of the transit time (which instead depends on a linear way from the radius r). Figure 9a–c show that the transverse collection configuration is able to work above 1 GHz if the radius r of the graphene active area is lower than 18 μm making harder the optical coupling with the incoming radiation.

Finally, Figure 9d shows the device NEP for DBRs composed by 3, 4 and 5 Si_3_N_4_/a-Si:H pairs. NEP has been calculated by Equations (5) and (6) (with A^*^ = 32 A/cm^2^K^2^, *T* = 300 K, *k* = 8.617 × 10^−5^ eV/K and *qΦ*_B0_ = 0.45 eV) and by taking into account the results shown in Figure 8a. NEP decreases by increasing the finesse of the cavity due to the increase in responsivity, the minimum NEP at 1550 nm is 0.6 $W/\mathrm{cm}\sqrt{\mathrm{Hz}}$ for a DBR with 5 Si_3_N_4_/a-Si:H pairs.

### 4.2. Longitudinal Collection Configuration (LCC)

In this section we investigate the longitudinal collection configuration (LCC) shown in Figure 6b. As output mirror, a DBR constituted of 3, 4 and 5 Si_3_N_4_/a-Si:H pairs is considered. 

On the other hand, the input mirror of the device is constituted by a DSOI-HD. Of course, as also shown in Figure 6a, the optical microcavity is formed by a-Si:H/graphene/c-Si three-layer structure. 

We have performed numerical simulations in order to calculate the graphene absorption at 1550 nm by varying the thicknesses of both a-Si:H and c-Si layers comprising the cavity for DBRs with 3, 4 and 5 Si_3_N_4_/a-Si:H pairs. The maximum graphene absorption can be achieved when the thickness of c-Si and a-Si:H are 114.9 nm and 214.0 nm, respectively. The spectral graphene absorption around 1550 nm for these optimized thicknesses is shown in Figure 10a for DBRs constituted of 3, 4 and 5 Si_3_N_4_/a-Si:H pairs. 

Figure 10a shows that the maximum graphene absorption is 0.24, 0.28 and 0.29 while the FWHM are 13.42 nm, 12.43 nm and 12.13 nm, for DBRs with 3, 4 and 5 Si_3_N_4_/a-Si:H pairs, respectively. By contrast the DSOI mirror, due to its first heavily doped c-Si layer, is characterized by a free carrier absorption of 0.53, 0.62 and 0.66 for DBRs with 3, 4 and 5 Si_3_N_4_/a-Si:H pairs, respectively. For this reason, the graphene optical absorption is lower than that one reported for transverse collection configuration. 

Even if Figure 3b,d show that the absorption coefficient of graphene is higher than heavily doped c-Si, the latter is much thicker (340 nm-thick) with respect to graphene (0.335 nm-thick), thus absorbing the most part of the light trapped into the cavity. By applying Equations (1) and (2) we can achieve the spectral responsivity at zero bias. As shown in Figure 10c the maximum responsivity at 1550 nm is 0.10 A/W, 0.12 A/W and 0.13 A/W for DBRs with 3, 4 and 5 Si_3_N_4_/a-Si:H pairs, respectively. In order to verify if a further increase in responsivity at 1550 nm can be obtained by increasing the reverse bias we use Equations (1)–(3), also in this case the increase in responsivity at −10 V is very limited, as reported in Figure 10d. 

Moving our attention to the bandwidth of the device, the time constants discussed in Section 2 have been calculated as already described for TCC. The only difference concerns the calculation of both the transit time *τ_tr_* = *t*/vsat, being t the thickness of the c-Si layer composing the cavity (*t* = 114.9 nm) and the cavity photon lifetime *τ_ph_* = 1/2*πδν* which is expected to reduce due to the increase in *δν* associated to the increased cavity losses. The frequency spectral widths *δν* have been calculated by Figure 10a as 1675, 1552 and 1515 GHz for DBR with 3, 4 and 5 Si_3_N_4_/a-Si:H pairs, respectively. Figure 11a shows three time constants and 3 dB roll-off frequency for a device provided of a DBR constituted by 5 Si_3_N_4_/a-Si:H pairs. Due to the reduced transit time in LCC, Figure 11a shows as the limiting factor is now the RC time constant. Because the junction capacity C is linked to the graphene area in contact with Si, by reducing the area an increase in bandwidth is expected, on the other hand, a smaller area could make harder the optical coupling of the incoming radiation. It is worth noting that in this configuration not only the cavity photon lifetime but also the transit time are independent of the radius r of the graphene active area. Figure 11a shows that if the radius of the active area is 70 μm the LCC is able to work at 1 GHz while at the same radius the TCC is characterized by a bandwidth of only 186 MHz (see Figure 9c).

Finally, Figure 11b shows the device NEP for DBRs composed of 3, 4 and 5 Si_3_N_4_/a-Si:H pairs. NEP has been calculated by Equations (5) and (6) as already discussed for the TCC. NEP decreases by increasing the finesse of the cavity due to the increase in responsivity, the minimum NEP at 1550 nm is 1.26 $W/\mathrm{cm}\sqrt{\mathrm{Hz}}$ for a DBR with 5 pairs of Si_3_N_4_/a-Si:H.

## 5. Discussion

In this section we put in comparison the optimized transverse and longitudinal collection Fabry–Pérot graphene/Si Schottky PD. As the output mirror is not only considered a DBR constituted by Si_3_N_4_/a-Si:H (5 pairs for optimized structures) but also a 200 nm-thick gold Au MR which could be a good option for reducing the manufacturing complexity. However, the metallic mirror absorbs part of the light trapped in the cavity at any round-trip reducing the graphene absorption, consequently a reduced responsivity is expected with respect to the counterpart based on DBR.

Figure 12a shows a comparison of the spectral graphene absorption for the four structures (transverse and longitudinal collection configuration with both DBR and MR as output mirror); as expected, the maximum graphene absorption is obtained for the configuration which does not involve other absorbing layers apart from graphene. As a consequence, Figure 12b shows that the TCC is characterized by the highest responsivity of 0.24 A/W which is a very interesting value mainly by considering that these Si-based PDs could be monolithically integrated with an electronic circuitry and not separately fabricated and then assembled as happened for the fabrication of NIR imaging systems based on InGaAs or germanium. As shown in Figure 12d, the same configuration is also characterized by the lowest NEP of 0.6 nW/cm $\sqrt{\mathrm{Hz}}$ and this is why this configuration could be preferred for applications where high sensitivities are required, for instance in free space optical communications in both spatial and terrestrial environment.

By contrast, Figure 12c shows as the longitudinal configuration is characterized by higher bandwidth than transverse counterpart. This is due to the reduced transit time which make the RC time constant the limiting factor. In LCC, a further increase in bandwidth could be obtained by reducing the RC time constant, for instance by reducing the graphene area in contact with silicon, i.e., by reducing the radius r. However, it is worth mentioning that if the active area becomes too small more complex optical coupling techniques are required for focusing the radiation on the active area. For a radius r = 70 μm, longitudinal structures provided by both DBR and MR output mirrors, are characterized by a bandwidth of 1 GHz while the transverse one by a bandwidth of only 186 MHz. LCC could be used for applications where the high speed is the main requirement. 

Figure 12a,b,d show that from a point of view of graphene absorption, responsivity and NEP, the LCC provided of DBR as output mirror is almost equivalent to the TCC provided of MR as output mirror, thus for applications where high bandwidth is not the main requirement, the transverse structure could be preferred because characterized by a lower manufacturing complexity. 

The performance of any configuration and related optimization parameters are reported in Table 2. Table 2 shows that concerning the structures provided of DBR as output mirror, the longitudinal configuration is characterized by the highest responsivity × bandwidth parameter, while the transversal one is characterized by the narrowest FWHM, i.e., by the highest selectivity. By contrast, the lowest selectivity is shown by the longitudinal configuration provided of MR as output mirror due to the highest losses in the cavity given by both the metal reflector and the doped DSOI.

## 6. Conclusions

In this work we have theoretically investigated the performance of a new concept of near-infrared Fabry–Pérot graphene/silicon Schottky photodetector based on a double silicon on insulator substrate. The absorption mechanism, based on the internal photoemission effect, can be enhanced by exploiting the interference phenomena inside the optical microcavity. All numerical simulations have been carried out by the transfer matrix method and taking into account the physics behind the hot carrier emission from two-dimensional materials (graphene) into silicon. Moreover, for more accurate investigation, dispersion of all materials involved in the proposed structure have been taken into account, too. 

We have investigated and compared two configurations: one where the current is collected in the transverse direction with respect to the direction of the incident light, the other where it is collected in the longitudinal direction. We prove that while the TCC is characterized by the highest graphene absorption, highest responsivity and lowest NEP, the LCC is characterized by the highest bandwidth and responsivity × bandwidth product. Our results show responsivity of 0.24 A/W, bandwidth in the GHz regime and noise equivalent power of 0.6 nW/cm√Hz. In addition, the devices show a spectral selectivity which could be tuned with a proper choice of the cavity thickness. In this work TCC is characterized by a best selectivity of 8.5 nm (FWHM) around 1550 nm. 

A further increase in selectivity could be obtained by taking advantage of resonant structures characterized by higher-quality factors, moreover, thanks to the graphene broadband optical absorption these devices show the potentialities to work also at different wavelengths by simply changing the length of the three-layer cavity. The whole structure has been conceived to be compatible with silicon technology and we believe that it could have a huge impact in the field of silicon photonics. Of course, for a full CMOS compatibility some challenges need to be first addressed, among them: the transferring of large-area graphene preserving a reasonable mobility, the low-resistance interconnection with graphene during the back-end-of-line (BEOL) process and the choice of suitable dielectric and encapsulation schemes for hysteresis-free and low-voltage operations.

## Figures and Tables

**Figure 1 micromachines-11-00708-f001:**
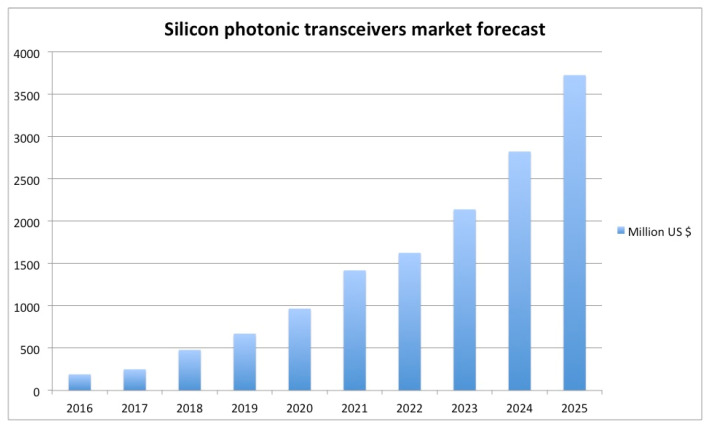
Silicon photonics 2016–2025 market forecast [1].

**Figure 2 micromachines-11-00708-f002:**
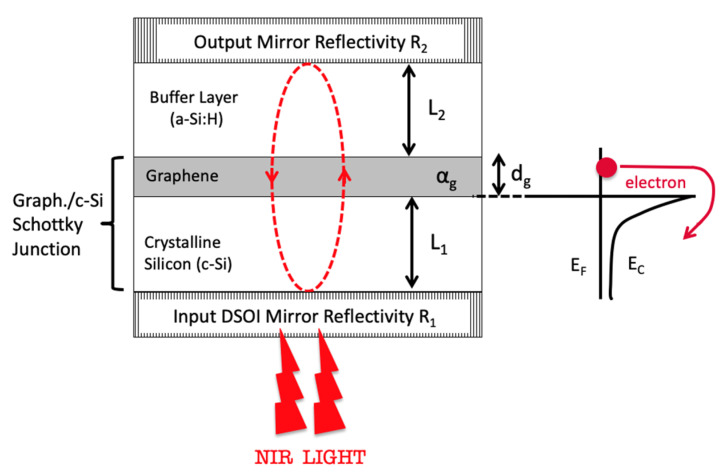
Simplified model of the proposed Fabry–Pérot graphene/silicon Schottky photodetector.

**Figure 3 micromachines-11-00708-f003:**
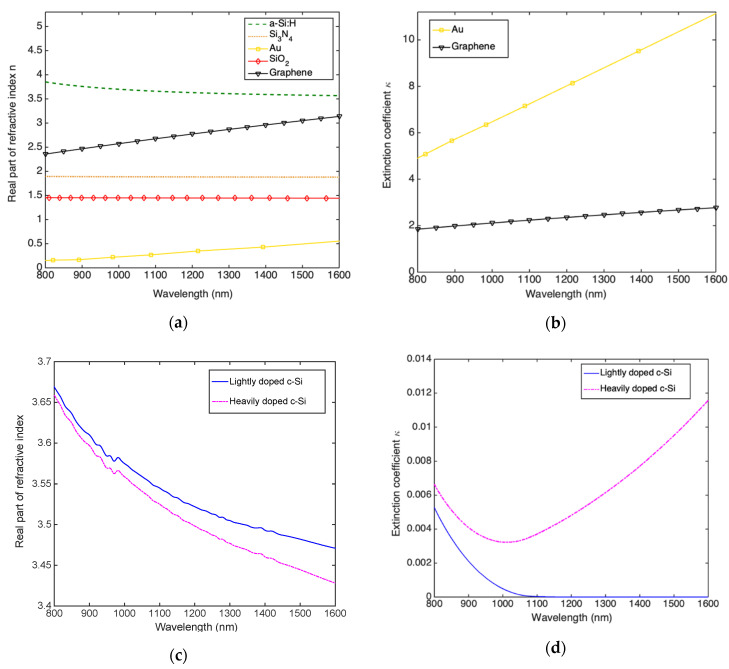
Dispersion curves of all materials used in our numerical simulation: (**a**) real part of refractive index and (**b**) extinction coefficient of a-Si:H, Si_3_N_4_, Au, SiO_2_ and graphene. (**c**) real part of refractive index and (**d**) extinction coefficient of lightly (1 × 10^15^ cm^−3^) and heavily doped (5 × 10^19^ cm^−3^) c-Si.

**Figure 4 micromachines-11-00708-f004:**
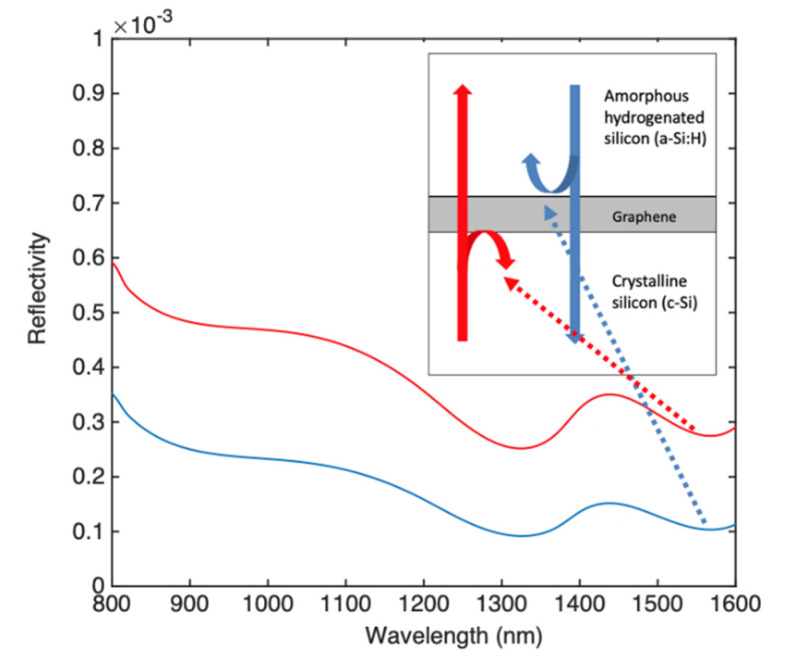
Reflectivity inside the cavity at the c-Si/graphene (red) and a-Si:H/graphene (blue) interfaces.

**Figure 5 micromachines-11-00708-f005:**
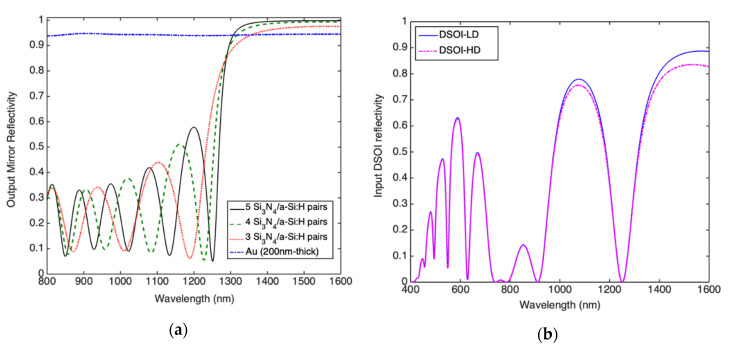
Reflectivity vs wavelength of the: (**a**) output mirror realized by 200 nm-thick Au metal reflector (MR) and distributed Bragg reflector (DBR) constituted by 3, 4 and 5 Si_3_N_4_/a-Si:H pairs and (**b**) input double silicon on insulator (DSOI) mirror constituted by the first c-Si layer both lightly and heavily doped.

**Figure 6 micromachines-11-00708-f006:**
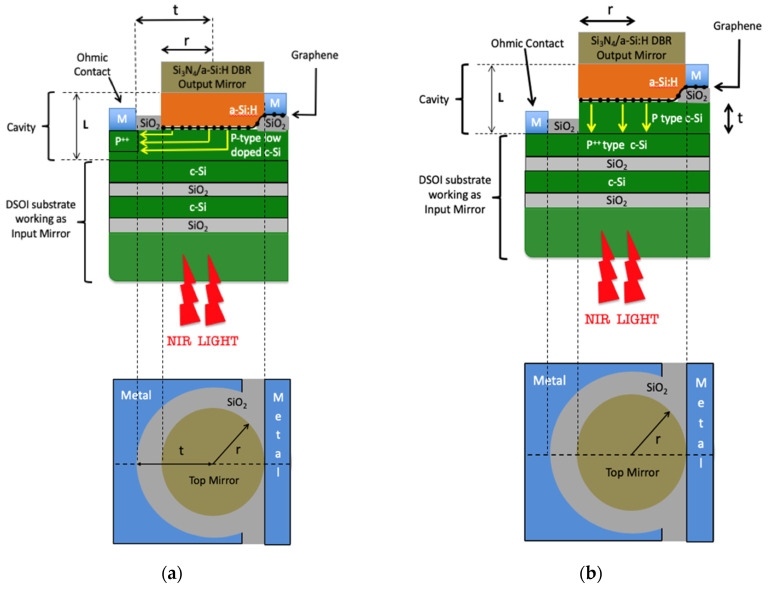
Sketch of the Fabry–Pérot graphene/Si Schottky PD in the (**a**) transverse and (**b**) longitudinal collection configuration.

**Figure 7 micromachines-11-00708-f007:**
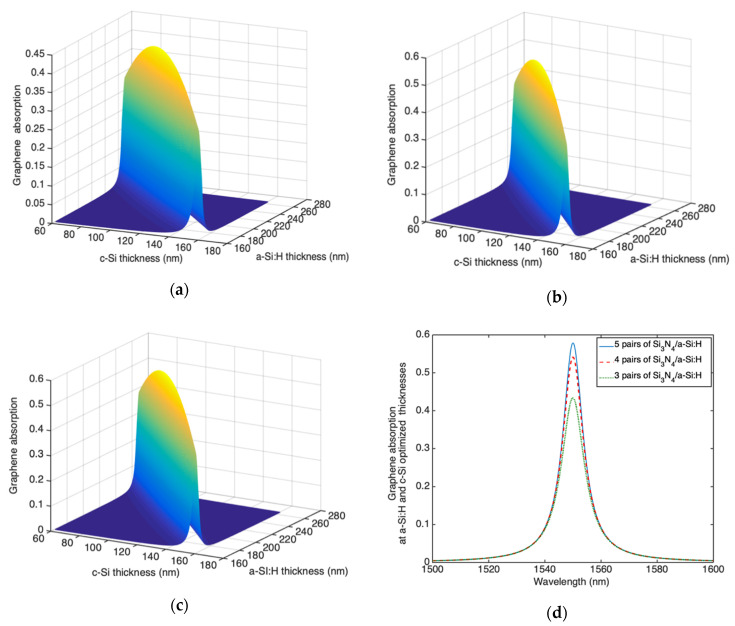
Graphene absorption as function of both c-Si and a-Si:H thicknesses for the transverse collection configuration (TCC) Fabry–Pérot graphene/Si Schottky PD provided of a DBR output mirror constituted by (**a**) 3, (**b**) 4 and (**c**) 5 pairs of Si_3_N_4_/a-Si:H and (**d**) spectral graphene absorption at the optimized c-Si and a-Si:H thicknesses.

**Figure 8 micromachines-11-00708-f008:**
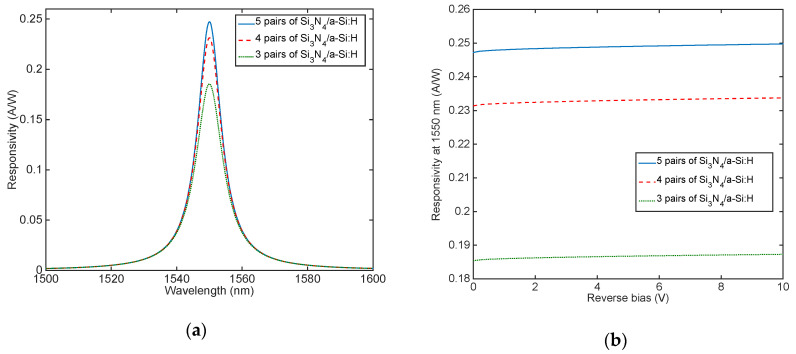
Responsivity (**a**) as function of the wavelength and (**b**) at 1550 nm as function of the reverse voltage applied for the TCC Fabry–Pérot graphene/Si Schottky PD.

**Figure 9 micromachines-11-00708-f009:**
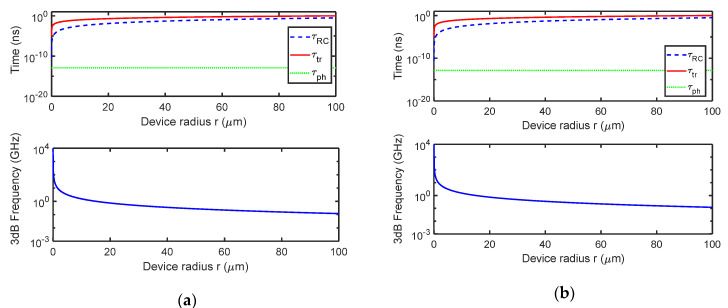
Carriers transit time *τ_tr_*, charge/discharge time *τ_RC_*, cavity photon lifetime *τ_ph_* and 3 dB roll-off frequency versus graphene disk radius r for the TCC Fabry-Pérot graphene/Si Schottky PD provided of a DBR output mirror constituted by (**a**) 3, (**b**) 4 and (**c**) 5 pairs of Si_3_N_4_/a-Si:H and (**d**) spectral noise equivalent power (NEP).

**Figure 10 micromachines-11-00708-f010:**
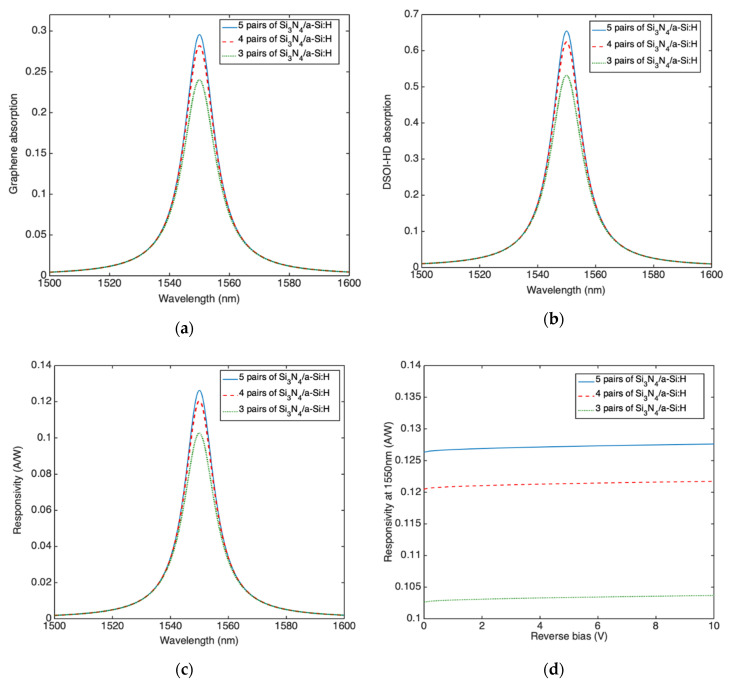
(**a**) Graphene optical absorption, (**b**) DSOI optical absorption, (**c**) responsivity as function of the wavelength for the longitudinal collection configuration (LCC) Fabry–Pérot graphene/Si Schottky PD provided of a DBR output mirror constituted by 3, 4 and 5 pairs of Si_3_N_4_/a-Si:H and (**d**) responsivity at 1550 nm as function of the reverse bias.

**Figure 11 micromachines-11-00708-f011:**
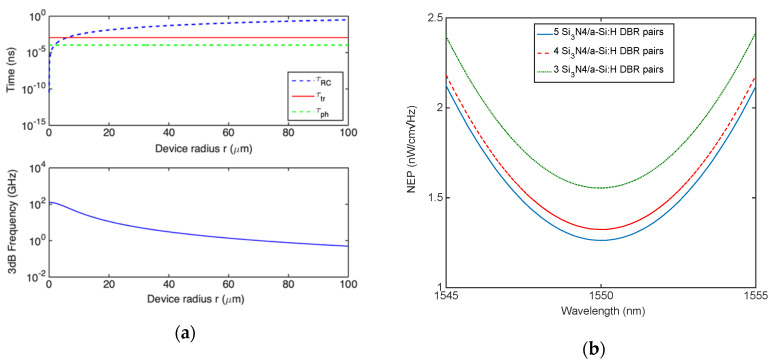
(**a**) Carriers transit time *τ_tr_*, charge/discharge time *τ_RC_*, cavity photon lifetime *τ_ph_* and 3 dB roll-off frequency versus graphene disk radius r for the LCC Fabry–Pérot graphene/Si Schottky PD provided of a DBR output mirror constituted by 5 pairs of Si_3_N_4_/a-Si:H and (**b**) spectral NEP.

**Figure 12 micromachines-11-00708-f012:**
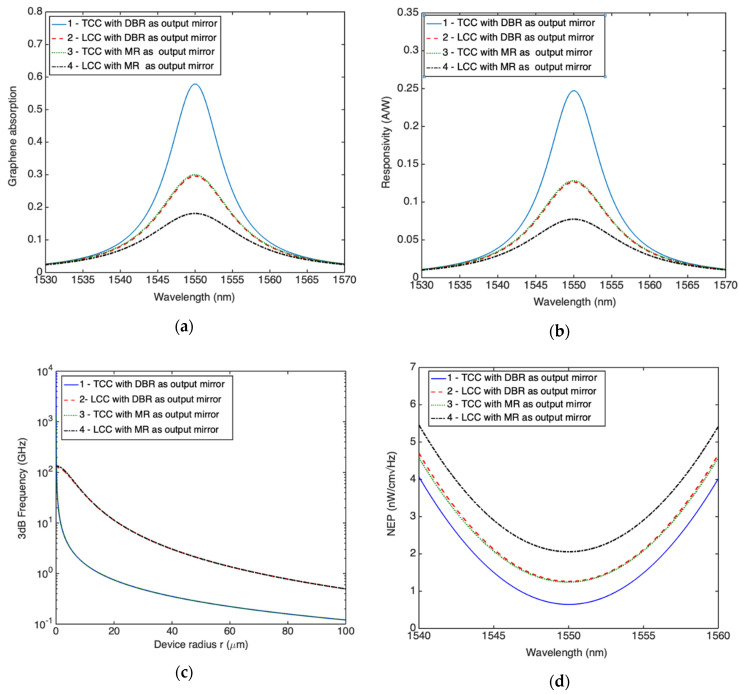
Comparison between optimized TCC and LCC Fabry-Pérot graphene/Si Schottky PDs provided of both DBR and MR as output mirror in term of: (**a**) spectral graphene absorption, (**b**) spectral responsivity, (**c**) 3 dB frequency and (**d**) spectral NEP.

**Table 1 micromachines-11-00708-t001:** Reflectivity and thicknesses of all investigated reflectors.

Reflector	Mirror	Reflectivity at 1550 nm	Thickness
DSOI-LD	Input	0.8790	340 nm (Si) and 270 nm (SiO_2_)
DSOI-HD	Input	0.8344	340 nm (Si) and 270 nm (SiO_2_)
DBR (3 pairs of Si_3_N_4_/a-Si:H)	Output	0.9756	213 nm (Si_3_N_4_) and 108 nm (a-Si:H)
DBR (4 pairs of Si_3_N_4_/a-Si:H)	Output	0.9932	213 nm (Si_3_N_4_) and 108 nm (a-Si:H)
DBR (5 pairs of Si_3_N_4_/a-Si:H)	Output	0.9985	213 nm (Si_3_N_4_) and 108 nm (a-Si:H)
Au	Output	0.9451	200 nm

**Table 2 micromachines-11-00708-t002:** Summary of the main performance of the optimized TCC and LCC Fabry–Pérot graphene/Si Schottky PD provided by both DBR and MR as output mirror.

Input Mirror	Output Mirror	c-Si Thick nm	a-Si:H Thick nm	FWHM nm	Resp. at 1550 nm A/W	3 dB Freq. at r = 70 μm MHz	Resp. × 3 dB Freq. A/WxMHz	NEP at 1550 nm nW/cm√Hz
1-TCC DSOI-LD	DBR (5 pairs)	111.0	214.0	8.54	0.24	186	44.6	0.60
2-LCC DSOI-HD	DBR (5 pairs)	114.9	214.0	12.13	0.13	1000	126.0	1.26
3-TCC DSOI-LD	Metal	110.1	303.7	12.18	0.13	186	23.8	1.24
4-LCC DSOI-HD	Metal	110.1	307.4	15.59	0.0775	1000	77.5	2.10

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
