# Peer review of "Theoretical Investigation of Near-Infrared Fabry–Pérot Microcavity Graphene/Silicon Schottky Photodetectors Based on Double Silicon on Insulator Substrates"

_micromachines, 2020, doi:10.3390/mi11080708_

Round 1

Reviewer 1 Report

This manuscript describes a theoretical investigation of near-infrared Fabry-Pérot microcavity graphene/silicon Schottky photodetectors. The content of the manuscript describes a depth discussion regarding the physics mechanism. The results show responsivity of 0.24A/W, bandwidth in GHz regime, and FWHM of 8.5nm. 

I suggest the following comments need to be addressed before this manuscript can be published in Micromachines. 

(1) Application limitation, if any: does this method requires particular fabrication tolerances in DBRs

-What are the limiting factors for the bandwidth? 

-How about the backside of wafer and limitation of responsivity due to the back reflection from the front (incident) DBR?

-How the peck wavelength and 8.5nm spectral bandwidth can be engineered to cover different applications? 

-Can this method be used for high-volume productions and if yes can you address the possible issues to be addressed.

(2) The results show responsivity of 0.24A/W and bandwidth in GHz. Are they fundamental or design-related?

(3) Any comments on the (possibly extra) technological challenges of this solution compared to other solutions (e.g., plasmatic and Ge)?

(4) Could such an approach be implemented for waveguide integrated graphene/silicon Schottky photodiodes?

Author Response

Response to Reviewer 1 is in the uploaded file.

Reviewer 2 Report

The author in "Theoretical investigation of near-infrared Fabry-Pérot microcavity graphene/silicon Schottky photodetectors based on double silicon on insulator substrates" presents a new concept of Si-based photodetector. The article is well written and the results are really interesting, and because of that, I recommend the publication after minor comments.

1) In the introduction, the author discussed the advantages and the state of arts of the Gr/Si photodetector. Inline 77, the author concludes his consideration of why graphene should be used to replace the metal. I suggest to strength this affirmation by citing works about Gr/Si photodetector such as:

doi.org/10.1088/2053-1583/aa6aa0

doi.org/10.3390/nano7070158

doi.org/10.3390/nano9050659

2) Please check the label in Figure 9 a), b), c)

Author Response

Response to Reviewer 2 is in the uploaded file.
